# Conventional Diagnostic Approaches to Dermatophytosis: Insights from a Three-Year Survey at a Public Dermatology Institute in Italy (2019–2021)

**DOI:** 10.3390/diagnostics15172245

**Published:** 2025-09-04

**Authors:** Eugenia Giuliani, Maria Gabriella Donà, Amalia Giglio, Elva Abril, Francesca Sperati, Fulvia Pimpinelli, Alessandra Latini

**Affiliations:** 1STI/HIV Unit, San Gallicano Dermatological Institute IRCCS, 00144 Rome, Italy; eugenia.giuliani@ifo.it (E.G.); alessandra.latini@ifo.it (A.L.); 2Microbiology and Virology Unit, San Gallicano Dermatological Institute IRCCS, 00144 Rome, Italy; amalia.giglio@ifo.it (A.G.); elva.abril@ifo.it (E.A.); fulvia.pimpinelli@ifo.it (F.P.); 3UOSD Clinical Trial Center, Biostatistics and Bioinformatics, Scientific Direction, San Gallicano Dermatological Institute IRCCS, 00144 Rome, Italy; francesca.sperati@ifo.it

**Keywords:** dermatomycoses, dermatophytosis, fungi, microscopy, fungal culture, skin, *Trichophyton*, tinea, diagnostic tests, routine

## Abstract

**Background/Objectives**: Dermatophytosis is a widespread superficial fungal infection affecting skin, hair, and nails. Its diagnosis is often based on conventional methods such as microscopy and fungal culture. Laboratory confirmation is essential for guiding appropriate treatment and preventing the misuse of antifungal agents, which can contribute to the emergence of antifungal resistance. We retrospectively assessed the burden and species distribution of dermatophytosis in individuals attending a public dermatology institute in Italy over a 3-year period (2019–2021). **Methods**: We analyzed 3208 samples from 3037 individuals with clinical suspicion of superficial mycosis. All samples underwent direct microscopic examination and fungal culture. Data were stratified by demographics, body site, and fungal species. Agreement between diagnostic methods was assessed using raw concordance and Cohen’s Kappa statistic. **Results**: Dermatophytes were confirmed in 667 samples (20.8%). Buttocks and genitals showed the highest positivity rates (37.5% and 36.4%, respectively). *T. rubrum* (56.8%) and *T. mentagrophytes* (30.7%) were the predominant species among the dermatophyte-positive specimens. Agreement between microscopy and culture was good (raw concordance: 91.6%, Cohen’s Kappa: 0.77, 95% CI: 0.74–0.79). Younger age and male gender were significantly associated with dermatophyte positivity. **Conclusions**: Our data provide updated epidemiological insights into dermatophytosis in Italy and support appropriate antifungal stewardship. Laboratory confirmation remains essential for an accurate diagnosis and species identification, thus avoiding other non-dermatophytic or non-infectious conditions being treated as dermatophytosis.

## 1. Introduction

Dermatophytosis is a benign superficial infection caused by filamentous fungi, called dermatophytes, which exhibit tropism for keratinized substrates such as skin, hair, and nails [1,2,3]. In Western countries, dermatophytes are responsible for almost all onychomycoses [2,4,5]. Dermatophytosis accounts for half of the estimated 650 million fungal skin infections globally [6]. Clinical manifestations of dermatophytosis, generally named “tinea”, are classified according to the affected body site [2]. These conditions include the athlete’s foot, *intertrigo*, mycosis of the nails with consequent nail dystrophies, and transient alopecia of the beard and scalp.

Data regarding the epidemiology of dermatophytosis have highlighted the great variability in their prevalence in different areas of the world [2,3,7,8,9,10,11], likely determined by several factors, including geographical and climatic conditions. Since dermatophytes preferentially develop in humid and warm conditions, their prevalence is generally higher in tropical and subtropical regions. Other determinants of dermatophytosis are migration flows, socio-economic conditions, lifestyle, co-morbidities, gender, age, and genetic predisposition [3,12,13]. Dermatophytosis is also a zoonotic infection that can affect domestic animals, which could contribute to an increased risk of infection in humans [13,14].

Although dermatophytosis is successfully treated in most cases, healthcare professionals are increasingly facing several major challenges, due to the limited range and availability of antifungal drugs, the prolonged treatment duration together with the antifungal resistance. The incorrect use of antimycotic drugs has led to the emergence of resistant dermatophytes, making the treatment more demanding [1,3,15]. Several resistance mechanisms have been described for the *Trichophyton* genus [16]. Remarkably, the recent years have also seen the emergence of new resistant strains [17], such as *Trichophyton indotineae*, a dermatophyte highly prevalent in India, responsible for chronic or recalcitrant infections [18].

Notably, suspected superficial mycosis are often treated empirically, a behavior that may lead to the emergence of resistance [17]. A laboratory-confirmed diagnosis is needed in order to avoid the unnecessary use of antifungal drugs and to adopt the correct treatment of a confirmed infection. Conventional diagnostic tests, i.e., microscopy and fungal culture, still represent the most widely used methods [19]. The direct microscopic examination is cheap and rapid, allowing for a prompt treatment initiation (pending the culture test result). Nonetheless, novel tools are emerging in the field of mycological diagnosis, such as nucleic acid amplification tests (NAATs) and matrix-assisted laser desorption ionization-time of flight mass spectrometry (MALDI-TOF MS) [17].

In view of the relevance of dermatophytosis and in order to obtain data on their epidemiological situation, we retrospectively analyzed the diagnoses performed over a 3-year period at the only public dermatological and research institute in Italy, which has extensive experience in dermatological infections and is able to provide a comprehensive and informative retrospective case history. Notably, our institute includes an outpatient clinic with staff skilled in the collection of clinical samples from individuals with a suspicion of superficial mycosis and experienced in the interpretation of conventional mycological diagnostics.

## 2. Materials and Methods

### 2.1. Study Population and Data Collection

The records of the Microbiological Sampling Clinic, belonging to the Microbiology and Virology Laboratory of the San Gallicano Dermatological Institute (Rome, Italy), were retrospectively reviewed (January 2019–December 2021). The study included all the individuals who satisfied the following criteria: (i) had clinical features suggesting a superficial mycosis; (ii) were subjected to a skin/hair/nail sampling, according to the clinical suspicion; (iii) had a result for the direct microscopic examination as well as for the fungal culture. Information about socio-demographic (age, gender) data, as well as microscopic and culture examination results, were retrieved from the records.

### 2.2. Microscopy and Culture Test

The direct mycological examination was performed by light microscopy using 20% potassium hydroxide (KOH) with 40% dimethyl sulfoxide (DMSO) mount (Santa Cruz Biotechnology Inc., Heidelberg, Germany) on the slide with the clinical sample. Depending on whether the sample was collected from the skin or nails, the slide was observed after 1–4 h for the research of fungal spores and hyphae fragments. The fungal culture, always performed in parallel with the direct mycological examination, used Sabouraud Dextrose Agar (SDA) +/− cycloheximide (Becton Dickinson, Milan, Italy), a culture medium that ensures optimal growth of mycetes limiting the possibility of contamination by bacteria and/or molds. Dermatophytes were identified at the species level based on the macro-morphological characteristics of the colonies after 7–15 days of incubation at 27 °C. A sample was considered dermatophyte-positive when dermatophyte growth was observed in the culture, regardless of the microscopic examination result. Both the microscopic evaluation and interpretation of the fungal culture were performed by dermatologists with 10 to 30 years of expertise in the field of mycological diagnostics.

### 2.3. Statistical Analysis

The prevalence of dermatophytosis, overall, according to the anatomical site, and the etiological agent (i.e., the different dermatophyte species belonging to the genera *Trichophyton*, *Microsporum,* and *Epidermophyton*) was estimated both for the entire study population and after stratification according to gender and nationality. The categorical variables were reported through absolute and relative frequencies, whereas the continuous variables through medians and interquartile range (IQR). Kolmogorov–Smirnov normality test was calculated for the continuous variables. To explore the differences in continuous variables, Kruskal–Wallis test or Mann–Whitney test was applied, as appropriate. The associations between categorical variables were analyzed using the Chi-square (χ^2^) test. The raw agreement between the microscopic examination and the culture test result (any mycosis) was estimated. Cohen’s Kappa statistic (K) with its relative 95% confidence interval (95% CI) was used to assess concordance and was interpreted in a qualitative manner based on the Landis and Koch classification criteria. A *p* value of <0.05 was interpreted as statistically significant. All statistical analyses were performed using SPSS statistical software version 29 (SPSS Inc., Chicago, IL, USA) and MedCalc^®^ Statistical Software version 23.0.9 (MedCalc Software Ltd., Ostend, Belgium).

## 3. Results

### 3.1. Study Population

During the study period, 3208 samples were collected from 3037 individuals with a suspicion of mycosis (1654 women, 54.5%). Out of these 3037 subjects, 2878 (94.8%) contributed only a single sample. The majority of subjects attended the Microbiology Sampling Clinic in 2019 (1331; 43.8%), followed by 801 (26.4%) in 2020, and 905 (29.8%) in 2021. The median age of the women was 55 years (IQR: 40–67) and that of men was 54 years (IQR: 36–68). The subjects were mostly Italian (2833, 93.3%). Figure 1 provides a detailed overview of the body areas sampled during the 3-year period. The majority of the samples were collected from the toenails (1356, 42.3%), followed by the feet (487, 15.2%).

### 3.2. Microscopic Examination and Culture Test

At microscopic examination, hyphae or other fungal structures were observed in 808/3208 samples (25.2%) (Table 1). Of the 808 microscopy-positive specimens, 585 (72.4%) tested positive for dermatophytes at the culture test, whereas 39 (4.8%) tested positive for fungi other than dermatophytes. Despite being negative at the microscopic examination, 82 additional samples were positive for dermatophytes at the culture test, giving a total of 667 ascertained cases of dermatophytosis out of the 3208 samples collected (20.8%). The raw concordance between the microscopic examination and the culture test was 91.6%, with a Cohen’s Kappa of 0.77 (95% CI: 0.74–0.79).

Although the number of accesses and samples collected during the pandemic period (2020 and 2021) was lower compared to the pre-pandemic year (2019), the dermatophyte positivity rate remained almost constant over the three years (296/1425, 20.8% in 2019; 158/845, 18.7% in 2020; 213/938, 22.7% in 2021).

Out of the 3037 individuals, 629 (20.7%) tested dermatophyte-positive in at least one sample, and the majority were men (369, 58.7%). The proportion of men testing positive for dermatophytes was significantly higher compared to women (369/1383, 26.7% vs. 260/1654, 15.7%; *p* < 0.001). The positivity rate among non-Italian individuals was significantly higher than that among Italians (54/204, 26.5% vs. 575/2833, 20.3%; *p* = 0.036). Dermatophyte-positive individuals were significantly younger than negative ones [50 years (IQR: 30–65) vs. 55 years (IQR: 39–68), respectively, *p* < 0.001]. This finding was confirmed also among women [48 years (IQR: 26–62) vs. 56 years (IQR: 43–67), respectively, *p* < 0.001] but not among men [53 years (IQR: 35–66) vs. 55 years (IQR: 36–68), respectively, *p* = 0.259].

Dermatophyte positivity by body site is shown in Table 2. Buttocks exhibited the highest prevalence of dermatophytosis (24/64, 37.5%), followed by genitals (64/176, 36.4%), and upper limbs (44/125, 35.2%).

### 3.3. Dermatophyte Species

The majority of the 667 confirmed diagnoses of dermatophytosis were caused by *T. rubrum*, which was responsible for 379 cases (56.8%), followed by *T. mentagrophytes* (205, 30.7%), and *Microsporum canis* (*M. canis*) (50, 7.5%) (Figure 2). The majority of the samples positive for *T. rubrum* were collected from the feet (109/379, 28.8%), while those positive for *T. mentagrophytes* were collected from the toenails (76/205, 37.1%). Most of the *M. canis* diagnoses were found in samples from the upper extremities (16/50, 32.0%).

Out of the 3208 samples, 42 (1.3%) tested positive for non-dermatophyte fungi; 40 for Candida spp. (16 for *C. albicans*, 2 for *C. lusitaniae*, 1 for *C. orthopsilosis*, 20 for *C. parapsilosis* and 1 for *C. tropicalis*), 1 for *Malassetia furfur* and 1 for *Scopulariopsis brevicaulis*.

## 4. Discussion

Dermatophytosis represents the most common fungal infection, affecting approximately a quarter of the global population, with variations depending on geographical regions [20,21]. An accurate diagnosis and the correct identification of the species causing the infection is essential for guiding appropriate treatment and avoiding misuse of antifungal drugs [22]. Microscopy and fungal culture are among the most widely used methods for diagnosing dermatophyte infections. In the present study, we observed a good concordance between these two methods. Although multiple samples from the same individuals were included in the concordance analysis, they represented only a small fraction of the total dataset, minimizing the risk of a biased estimate. Indeed, the agreement calculated only for individuals who contributed a single sample was almost identical to that estimated for the total dataset [Cohen’s Kappa of 0.76 (95% CI: 0.73–0.79)]. Microscopy–culture concordance remains variable [19,23,24], since it is influenced by several factors. Firstly, direct microscopy is less sensitive than culture, so that a negative result cannot exclude a fungal infection [19]. The expertise of the investigators in collecting and examining the clinical specimens also represents an influential aspect. Accurate diagnosis relies heavily on the quality of clinical sampling [19]. Inadequate or contaminated samples can lead to false negatives, misidentification of pathogens, and ultimately, inappropriate treatment. Equally important is the involvement of skilled personnel in the diagnostic pathway, since mycological diagnostics may be complex and require a well-consolidated expertise to interpret subtle variations in fungal morphology and growth characteristics. Both the investigators involved in this study had long-lasting experience in the field. One of them was a senior dermatologist with over 30 years of experience in collecting skin samples for mycological diagnosis and identifying dermatophytes based on their morphological characteristics.

Although conventional diagnostic tools are reliable and widely used, the slow growth of dermatophytes causes delayed pathogen identification. In addition, not all isolates produce morphological features that allow species identification. Other tools have thus emerged as potential aids in the diagnosis of dermatophytosis. Among these, molecular techniques based on Polymerase Chain Reaction (PCR) are increasingly used. Various methods have been developed over the last decade, such as conventional PCR, quantitative/real-time PCR (qPCR), nested PCR, multiplex PCR, PCR and Enzyme-linked immunosorbent assay (PCR-ELISA) and PCR-Restriction fragment length polymorphism (PCR-RFLP) [25,26]. These methods differ in terms of processing time, technical complexity, accuracy and range of detectable species [26]. Conventional PCR is a widely used, cost-effective method for detecting dermatophytes with high sensitivity, although it requires post-amplification steps. A higher sensitivity and specificity is offered by qPCR, which is useful for distinguishing infection from contamination. Multiplex PCR allows detection of multiple pathogens simultaneously with good accuracy. Hybrid methods, such as PCR-ELISA and PCR-RFLP, enhance sensitivity but are time-consuming and not commonly applied [25,26]. Despite PCR-based methods offer several advantages over conventional diagnostics, such as rapid identification, the possibility to analyze simultaneously multiple samples and to differentiate closely related species, their high sensitivity may cause false-positive results. In addition, they are not suitable for monitoring the effectiveness of the treatment, since DNA from non-viable cells can be detected [22]. A DNA microarray has been recently developed for the accurate, rapid and automated detection of 13 fungal species [27]. The excellent performance and the high-throughput capability of this novel method make it a promising tool for routine diagnosis of dermatophytosis. MALDI-TOF MS, which is effectively used in routine bacterial identification, also represents a promising option in mycological diagnostics [17,28,29,30]. At the time of the sample evaluation, PCR and MALDI-TOF MS had not yet been implemented as routine methods in the Microbiology Laboratory. 

We observed that about one-fifth of the subjects referred to the Microbiological Sampling Clinic with a suspicion of superficial mycosis had dermatophytosis. This finding underscores the importance of laboratory confirmation for accurate diagnosis, as non-dermatophytic molds and non-infectious conditions can mimic the clinical presentation of dermatophytosis. In our case series, the dermatophyte positivity rate did not change throughout the study period, despite the lower number of individuals tested during the pandemic period, which was expected to change some behaviors/habits especially during the lockdowns.

In our entire case series, individuals diagnosed with dermatophytosis were significantly younger than negative subjects (median age of 50 vs. 55 years), a trend that was also confirmed among women. Dermatophytosis may occur at any age, with certain infections occurring more frequently among children (e.g., *tinea capitis*), adults (e.g., *tinea pedis*) or the elderly (e.g., onychomycosis) [3,12].

We found that *T. rubrum* was the most represented dermatophyte, as already described in Italy by others [31,32] and in line with global data [3,15]. An elegant review by Seebacher C et al. had already described *T. rubrum* as the most common dermatophyte since the 1950s in central and northern Europe, followed by *T. mentagrophytes* [33]. In line with the fact that *T. rubrum* was the most prevalent etiologic factor of dermatophytosis observed in our cases series, *tinea pedis* represented the most prevalent dermatophytosis. Our result differs in part from those described by other Italian studies, which found *tinea corporis* as the prevalent dermatophytosis, even up to 8-fold more common than *tinea pedis* [31,34].

We did not observe any cases of *T. indotineae* during the study period. However, it must be underlined that this species poses a significant challenge in terms of diagnosis, since it is hardly distinguishable from *T. interdigitale* and *T. mentagrophytes* [18]. These species share overlapping phenotypic characteristics, such as colony morphology, microscopic features, and growth patterns, which can make differentiation based solely on conventional mycological methods extremely difficult [17,18]. Accurate identification of *T. indotineae* often requires advanced molecular techniques, such as sequencing of the internal transcribed spacer (ITS) region, which may not be routinely available in all diagnostic laboratories [17]. It is thus important to emphasize that the absence of confirmed cases in our study does not reflect the epidemiological scenario in Italy. Indeed, *T. indotineae* has spread worldwide and few cases have been described in our country [35].

*M. canis* was one of the most prevalent dermatophytes detected in this survey, despite being a zoophilic species. In Europe, particularly in Mediterranean countries, the incidence of *M. canis* infection increased in the early 2000s, confirming the shift from a zoophilic to an anthropophilic species [4,13]. In Italy, Panasiti et al. even found that *M. canis* accounted for the majority of the cases of dermatophytosis diagnosed from 2002 to 2004 [34]. The epidemiology of human infections caused by zoophilic dermatophytes depends on the frequency and intensity of contact with animals. Domestic animals, such as cats and dogs, are frequent sources of dermatophytosis and cases of human-to-human transmission of zoophilic dermatophytes have even led to nosocomial outbreaks. Identifying pets as a source of infection in humans can help prevent recurrences or new infections, particularly in children, by treating affected animals and their environments appropriately. Dermatophytosis, which is not traditionally considered as a sexually transmitted infection, can also be acquired through sexual intercourse [36,37] because of the close and intimate contact during sex and the involvement of anatomical regions associated with the route of the infection. In fact, several cases of sexually transmitted *T. mentagrophytes* have been described among men who have sex with men [18].

Our study presents a few limitations. It reflects a single-center experience, which may restrict the generalizability of the findings. It provides epidemiological data only up to 2021, which may not reflect post-pandemic epidemiological trends. It only relies on conventional diagnostics, which may not be accurate when pathogen identification cannot solely rely on morphological characteristics. Data regarding possible resistance to antifungal drugs were not available.

## 5. Conclusions

In conclusion, our study showed that conventional mycological tests confirmed dermatophytosis in approximately one-fifth of the over 3000 subjects referred to the Microbiology Laboratory because of suspicious superficial skin mycosis. This implies that laboratory confirmation is essential to avoid other non-dermatophytic or non-infectious conditions being treated as dermatophytosis. *T. rubrum* was the most common species in the culture-positive samples.

Since dermatophytosis continues to represent a significant public health concern worldwide, the formation of skilled personnel capable of recognizing fungal pathogens, interpreting laboratory results, and applying appropriate diagnostic techniques is vital. Mycological diagnostics often require nuanced interpretation and familiarity with diverse fungal morphologies and growth, which underscores the need for specialized training and continuous professional development. Nonetheless, the implementation of novel molecular technologies is pivotal to improve diagnostic accuracy.

## Figures and Tables

**Figure 1 diagnostics-15-02245-f001:**
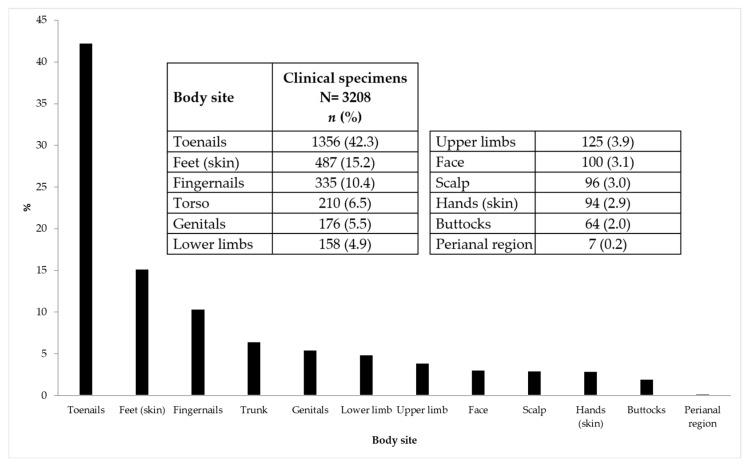
Overview of the body sites sampled for a suspected superficial mycosis at the Microbiological Sampling Clinic of the Microbiology and Virology Laboratory of the San Gallicano Dermatological Institute IRCCS (Rome, Italy) from January 2019 to December 2021 Absolute numbers and relative percentages are also shown (percentages do not add up to 100 due to rounding).

**Figure 2 diagnostics-15-02245-f002:**
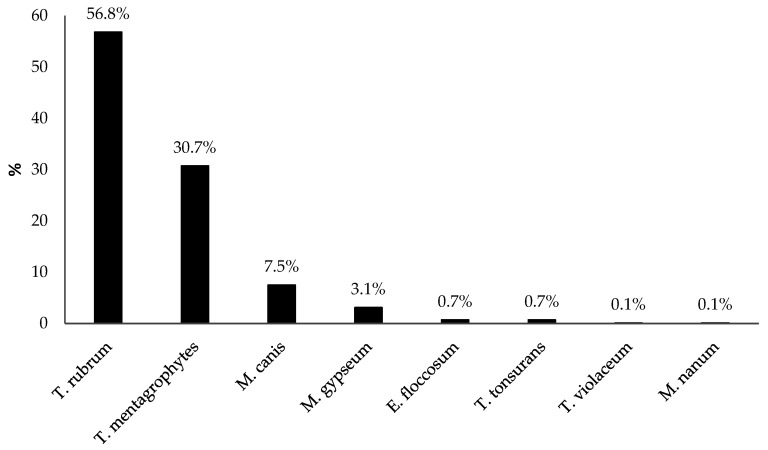
Dermatophyte species identified in the 667 samples that tested positive at the culture test.

**Table 1 diagnostics-15-02245-t001:** Results of microscopic examination and culture test for the 3208 clinical specimens analyzed during the study period.

Microscopic Examination	Fungal Culture Test, *n* (%)			
	Negative	Dermatophytes	Other Fungi	Total
Negative	2315 (96.5)	82 (3.4)	3 (0.1)	2400 (74.8)
Positive	184 (22.8)	585 (72.4)	39 (4.8)	808 (25.2)
Total	2499 (77.9)	667 (20.8)	42 (1.3)	3208 (100)

**Table 2 diagnostics-15-02245-t002:** Positivity for dermatophytes and non-dermatophytes-fungi (other fungi) at the culture test according to the body site.

Body Site	Dermatophytes *n* (%)	Other Fungi ^a^ *n* (%)
Toenails, *n* = 1356	216 (15.9)	4 (0.3)
Feet (skin), *n* = 487	163 (33.5)	1 (0.2)
Fingernails, *n* = 335	8 (2.4)	31 (9.3)
Torso, *n* = 210	34 (16.2)	1 (0.5)
Genitals, *n* = 176	64 (36.4)	4 (2.3)
Lower limbs, *n* = 158	41 (25.9)	0 (0.0)
Upper limbs, *n* = 125	44 (35.2)	0 (0.0)
Face, *n* = 100	26 (26.0)	0 (0.0)
Scalp, *n* = 96	25 (26.0)	0 (0.0)
Hands (skin), *n* = 94	20 (21.3)	1 (1.1)
Buttocks, *n* = 64	24 (37.5)	0 (0.0)
Perianal region, *n* = 7	2 (28.6)	0 (0.0)
Total, *n* = 3208	667 (20.8)	42 (1.3)

^a^ positivity for one of the following: *C. albicans*, *C. lusitaniae*, *C. orthopsilosis*, *C. parapsilosis*, *C. tropicalis*, *M. furfur*, *S. brevicaulis*.

## Data Availability

Dataset available on request from the authors.

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
