# Peer review of "Conventional Diagnostic Approaches to Dermatophytosis: Insights from a Three-Year Survey at a Public Dermatology Institute in Italy (2019–2021)"

_diagnostics, 2025, doi:10.3390/diagnostics15172245_

Round 1
Reviewer 1 Report
Comments and Suggestions for Authors
Dear Authors,
Thank you for your efforts. I have reviewed your article entitled “Conventional Diagnostic Approaches to Dermatophytosis: Insights from a Three-Year Survey at a Public Dermatology Institute in Italy (2019–2021)”
As you mentioned, microbiological diagnosis of dermatophytoses is important to avoid unnecessary antifungal treatment. This is why your article is valuable.
Although it is generally well designed and written, a few issues need to be clarified. My suggestions are as follows:
- Dermatophytoses? or Dermatophytosis? One of them should be preferred.
- In lines 191-192, a reference should be added to “Dermatophytoses are the most common fungal infections, affecting approximately a quarter of the global population, with variations depending on geographical regions.” statement.
- In lines, 199-206, references should be added to “Firstly, direct microscopy is less sensitive than culture, so that a negative result cannot exclude a fungal infection. The expertise of the investigators in collecting and examining the clinical specimens also represents an influential aspect. Accurate diagnosis relies heavily on the quality of clinical sampling. Inadequate or contaminated samples can lead to false negatives, misidentification of pathogens, and ultimately, inappropriate treatment. Equally important is the involvement of skilled personnel in the diagnostic pathway, since mycological diagnostics may be complex and requires a well-consolidated expertise to interpret subtle variations in fungal morphology and growth characteristics.” statement.
- In lines 221-222, a reference should be added to “However, since PCR can detect DNA from non-viable cells, it is not suitable for monitoring the effectiveness of the treatment and its high sensitivity may cause false-positive results.”statement.
- In lines 249-254, references should be added to “These species share overlapping phenotypic characteristics, such as colony morphology, microscopic features, and growth patterns, which can make differentiation based solely on conventional mycological methods extremely difficult. Accurate identification of T. indotineae often requires advanced molecular techniques, such as sequencing of the internal transcribed spacer (ITS) region, which may not be routinely available in all diagnostic laboratories.” statement.
Best regards,
Author Response
Dear Authors,
Thank you for your efforts. I have reviewed your article entitled “Conventional Diagnostic Approaches to Dermatophytosis: Insights from a Three-Year Survey at a Public Dermatology Institute in Italy (2019–2021)”
Comment 1: As you mentioned, microbiological diagnosis of dermatophytoses is important to avoid unnecessary antifungal treatment. This is why your article is valuable.
Response 1: thank you for appreciating our study.
Although it is generally well designed and written, a few issues need to be clarified. My suggestions are as follows:
Comment 2: Dermatophytoses? or Dermatophytosis? One of them should be preferred.
Response 2: we have opted for “Dermatophytosis” and replaced “Dermatophytoses” with “Dermatophytosis” throughout the text.
Comment 3: In lines 191-192, a reference should be added to “Dermatophytoses are the most common fungal infections, affecting approximately a quarter of the global population, with variations depending on geographical regions.” statement.
Response 3: we have now added two references for this statement (ref. #20 and #21) (page 6, line 197). The subsequent references have been renumbered consequently.
Comment 4: In lines, 199-206, references should be added to “Firstly, direct microscopy is less sensitive than culture, so that a negative result cannot exclude a fungal infection. The expertise of the investigators in collecting and examining the clinical specimens also represents an influential aspect. Accurate diagnosis relies heavily on the quality of clinical sampling. Inadequate or contaminated samples can lead to false negatives, misidentification of pathogens, and ultimately, inappropriate treatment. Equally important is the involvement of skilled personnel in the diagnostic pathway, since mycological diagnostics may be complex and requires a well-consolidated expertise to interpret subtle variations in fungal morphology and growth characteristics.” statement.
Response 4: These statements referred to the paper by Pihet et al, 2017 (ref. #19), already cited in the previous sentence. For clarity, we have now added again the ref. #19 at the end of the following sentences (page 6, lines 206-210).
Comment 5: In lines 221-222, a reference should be added to “However, since PCR can detect DNA from non-viable cells, it is not suitable for monitoring the effectiveness of the treatment and its high sensitivity may cause false-positive results.” statement.
Response 5: This statement referred to the paper by Heckler et al, 2023 (now ref. #22), already cited in the previous sentence. For clarity, we have now added again the ref. #22 at the end of the abovementioned statement (page 6, lines 229-231).
Comment 6: In lines 249-254, references should be added to “These species share overlapping phenotypic characteristics, such as colony morphology, microscopic features, and growth patterns, which can make differentiation based solely on conventional mycological methods extremely difficult. Accurate identification of T. indotineae often requires advanced molecular techniques, such as sequencing of the internal transcribed spacer (ITS) region, which may not be routinely available in all diagnostic laboratories.” statement.
Response 6: These statements referred to the papers by Gold and Lockhart, 2025 (ref. #17) and Jabet et al, 2023 (ref. #18), previously cited in the Introduction and Discussion. For clarity, we have now added again ref. #17 and #18 at the end of the abovementioned sentences (page 7, lines 262 and 265).
Reviewer 2 Report
Comments and Suggestions for Authors
Overall, this research describes accuracy of conventional diagnostic approaches to dermatophytosis and the frequency of species and body sites. The results support the notion that conventioanl diagostics are reliable. It could serve as a helpful reference for clinical practice with limitations clearly stated in the discussion.
Though, the authors claim that "molecular tools may further enhance diagnostic precision" is a discussion point not supported by clinical data from this article per se.
My additonal questions:
[Page 3, Line 123] According to the number of samples and individuals, most patients had only one sample to labs. Though I wonder if multiple samples submitted from single individuals, albeit small (171), could have skewed the results in any way. Because multiple samples from one person would likely have strong correlation in fungal species and microscropy-culture translatability.
[Page 6, Line 232] The authors suggest a trend of individuals diagnosed are younger than negative subjects. I wonder if this trend reflects the actual epidemiology because clinicians could be biased about which samples are sent for labs. For examples, physicians may more readily suspect dermatophytosis in older patients thus sending in more negative samples.
Author Response
Comment 1: Overall, this research describes accuracy of conventional diagnostic approaches to dermatophytosis and the frequency of species and body sites. The results support the notion that conventioanl diagostics are reliable. It could serve as a helpful reference for clinical practice with limitations clearly stated in the discussion.
Response 1: thank you for appreciating our study.
Comment 2: Though, the authors claim that "molecular tools may further enhance diagnostic precision" is a discussion point not supported by clinical data from this article per se.
Response 2: we have now rephrased the Conclusions of the Abstract to reflect more precisely what we can draw from the study results (page 1, lines 29-35).
My additonal questions:
Comment 3: [Page 3, Line 123] According to the number of samples and individuals, most patients had only one sample to labs. Though I wonder if multiple samples submitted from single individuals, albeit small (171), could have skewed the results in any way. Because multiple samples from one person would likely have strong correlation in fungal species and microscropy-culture translatability.
Response 3: thank you for pointing out this aspect. We have now specified that out of the 3037 individuals tested, 2878 (94.8%) contributed only one sample. The fact that only a minimal part of the subjects contributed multiple samples reduces the risk of a biased estimate. Indeed, the agreement calculated taking into account only the individuals with one sample is almost identical to the one calculated on the overall samples [Cohen Kappa 0.76 (95% CI: 0.73-0.79) vs. 0.77 (95% CI: 0.74-0.79)]. For this reason, we did not modify the data regarding the agreement, but we have now added this point to the Discussion (page 6, lines 201-205).
Comment 4: [Page 6, Line 232] The authors suggest a trend of individuals diagnosed are younger than negative subjects. I wonder if this trend reflects the actual epidemiology because clinicians could be biased about which samples are sent for labs. For examples, physicians may more readily suspect dermatophytosis in older patients thus sending in more negative samples.
Response 4: dermatophytosis is suspected on a clinical basis and may occur at any age; therefore, older age is not a criterion for sending samples for the mycological test. In addition, we found that dermatophyte-positive individuals had a median age of 50 vs. 55 years for the negative subjects, a significant, yet minimal difference. For sake of clarity, we have now rephrased this paragraph (page 6, lines 240-247).